

# Influence of organic, synthetic and biofertilizers on the diversity of cassava rhizosphere microbiome in Northeastern Thailand

Suthasinee Somyong[1], Wuttichai Mhuantong[2], Phakamas Phetchawang[1], Derrick Keith Thompson[3], Ornprapa Thepsilvisut[3] and Wirulda Pootakham[1]

[1] National Omics Center, National Center for Genetic Engineering and Biotechnology (BIOTEC), National Science and Technology Development Agency (NSTDA), Klong Luang, Pathum Thani, Thailand
[2] Enzyme Technology Research Team, Biorefinery Technology and Bioproduct Research Group, National Center for Genetic Engineering and Biotechnology, Klong Luang, Pathum Thani, Thailand
[3] Department of Agricultural Technology, Faculty of Science and Technology, Thammasat University Rangsit Center, Klong Luang, Pathum Thani, Thailand

Corresponding author
Wirulda Pootakham,
wirulda.poo@biotec.or.th

## ABSTRACT

Cassava, one of Thailand's main economic crops, is capable of growing in nearly all soil types. However, continuous monocropping depletes soil nutrients over time. Adopting good agricultural practices can help farmers reduce costs while improving soil fertility. The aim of this study was to compare cassava rhizosphere microbial communities resulting from cultivation under eight different fertilizer treatments, including synthetic, organic, and biological fertilizers, and to identify beneficial microbes that promote cassava growth and yield. The study was conducted at two sites in Northeastern Thailand. Results show that bacterial abundance and species richness (alpha diversity) peaked at 5 months after planting (MAP), showing a significant increase compared to 2 MAP. However, by 10 MAP, alpha diversity began to decline at both sites, Nampong and Seungsang. Among the treatments, the most notable differences in alpha diversity were observed at 5 MAP. At the Nampong site, experimental treatments with chicken manure (T3) and chicken manure combined with other fertilizers (T5, T6, and T8) exhibited significantly higher alpha diversity than did the control (without fertilizer, T1). At the Seungsang site, sole treatment with the full recommended rate of chicken manure (T3), and half of the recommended dose of synthetic fertilizer combined with half the recommended dose of chicken manure (T6) resulted in greater alpha diversity than did swine manure extract application (T4), half of the recommended dose of synthetic fertilizer combined with half of the recommended dose of swine manure extract (T7), and chicken manure application combined with stalk inoculation with plant growth-promoting rhizobacteria (PGPRs) (T8). Since T3 and T8 had the most significant impact on microbial abundance and diversity, as well as cassava growth and yield, the predominant bacteria in these treatments were identified as key targets. A total of eight target bacterial genera were identified: *Pseudomonas*, *Tumebacillus*, *Lysinibacillus*, *Paenibacillus*, *Dongia*, *Acidibacter*, *Sphingomonas*, and *Bacillus*.

Among them, *Tumebacillus* was the most notable, as it showed a significant correlation with fresh tuber yield. These beneficial bacteria may serve as key candidates for future biofertilizer production.

# INTRODUCTION

Cassava (*Manihot esculenta* Crantz) is one of the key economic plants that produce starch, which is used in the food and processing industries, particularly in animal feed production and ethanol production. It can be grown in almost all types of soil and adapt well to low-fertility soils. However, growing cassava as a monocrop for several years leads to the deterioration of soil nutrients. In addition to soil quality, several other factors contribute to cassava growth and production, including plant pests, abiotic factors (such as rainfall, environmental temperature, water and sunlight), agricultural management, and the addition of microorganisms in the form of fertilizers (*Compant et al., 2019*; *Prasetyo & Qomariyah, 2021*; *Omodara et al., 2023*). Microorganisms are on and within all plant crops, such as tomato, rice and cassava, in the phyllosphere, endosphere, and rhizosphere (*Dong et al., 2019*; *Frediansyah, 2021*; *Teheran-Sierra et al., 2021*; *Danso Ofori et al., 2024*). However, the rhizosphere microbiome has received the most attention in cassava research because it is primarily involved in tuber productivity (*Ha et al., 2021*; *Zeng et al., 2021*; *Huang et al., 2024*). The utilization of microorganisms can help increase plant productivity by enhancing soil fertility through the addition of nutrients, as well as improving disease and insect resistance (*Wang et al., 2022*; *Chaudhary et al., 2023*). Biotic interactions among microorganisms play an important role in supporting plant growth and production through several mechanisms, such as cell-to-cell diffusible signaling, contact-dependent interactions, secondary metabolites, metabolic interplay, and possibly other yet unknown mechanisms (*Singh et al., 2009*; *Venturi & Bez, 2021*). Good soil management, which involves increasing beneficial microorganisms in the rhizosphere of plants, can also be achieved by reducing the use of chemicals, including synthetic fertilizers, insecticides, and herbicides, as well as by practicing crop rotation and intercropping with other commercial species, like peanut and other legumes. (*Chen et al., 2020*; *Tang et al., 2020*; *Giacometti et al., 2021*; *Han, Dong & Zhang, 2021*). This helps to increase the diversity of microorganisms in the soil and enhance disease resistance. The addition of organic or biological fertilizers, directly to the soil, helps plants to gain benefit from beneficial microorganisms. Biofertilizers have advantages over organic fertilizers, such as chicken manure, swine manure, and others, because they deliver specific microorganism species in certain quantities. However, the application of both organic fertilizers and biofertilizers together may provide the greatest benefits, in terms of the diversity of beneficial microorganisms. Several studies have reported the benefits of using various beneficial

microorganisms in biofertilizers (*Abdel Aal et al., 2023*; *Shahwar et al., 2023*; *Dzvene & Chiduza, 2024*). For example, *Enterobacter agglomerans*, *Pseudomonas* spp., and *Achromobacter* spp., can solubilize phosphate, *Pantoea agglomerans*, *Burkholderia* spp., *Bacillus* spp., and *Psudomonas* spp. can produce IAA hormones that help plant growth, and *Paenibacillus* spp. and *Bacillus pumilus* can inhibit pathogenic fungi by producing surfactins and pumilacidins, respectively (*Frediansyah, 2021*). Microorganisms in the rhizosphere can be both beneficial and pathogenic to plants. They engage in symbiotic relationships as well as free-living interactions in the soil and they promote plant growth both directly and indirectly. Directly, they help decompose organic matter and plant residues into nutrients, aid in nitrogen fixation, phosphate solubilization, potassium solubilization, plant hormone production, and support root growth. Indirectly, they produce antibiotics that enhance resistance to pathogens, produce siderophores that bind iron and make it available for plant absorption, and contribute to tolerance to environmental stresses such as drought, salinity, and heavy metals (*Ojuederie & Babalola, 2017*; *Singh et al., 2023*; *Naz et al., 2024*). Moreover, studies have shown that combining biofertilizers with synthetic fertilizers can increase cassava yield while reducing the use of synthetic fertilizers by approximately 50% (*Otaiku, 2019*; *Wongsuwan, Khaengkhan & Sinsiri, 2021*). Synthetic fertilizers can alter the fungal community structure in cassava rhizosphere soil and impact yield (*Cai et al., 2021*). Therefore, replacing at least some portion of the applied synthetic fertilizer in cassava fields with application of biofertilizers has potential for maintaining healthy soil food web dynamics.

Currently, high-throughput sequencing plays a crucial role in studying the microbiomes of various major crops, such as rice (*Eyre et al., 2019*; *Zhao et al., 2020*; *Song et al., 2021*; *Wang et al., 2021*), wheat (*Eyre et al., 2019*; *Kavamura et al., 2021*), sugarcane (*Ishida et al., 2022*; *Malviya et al., 2022*), oil palm (*Berkelmann et al., 2020*; *Goh et al., 2020*) and cassava (*Cai et al., 2021*; *Alleyne, Mason & Vallès, 2023*). Studies of the cassava microbiome, including that in the phyllosphere, endosphere, and rhizosphere, have been conducted using 16S rRNA amplicon Illumina sequencing, since this method enables rapid analysis of its microbiome (*Frediansyah, 2021*; *Ha et al., 2021*; *Zeng et al., 2021*; *Zhang et al., 2021*; *Huang et al., 2024*).

Research on the effects of organic and biological fertilizer application on the cassava rhizosphere microbial community remains limited. The study aims to compare changes in microbial diversity and composition in the rhizosphere of cassava rhizosphere across different developmental stages, including the beginning, middle, and harvesting stages, under various soil management treatments. Using Illumina sequencing of 16S rRNA amplicons, we examine how different fertilizer types influence bacterial community profiles. The findings will help to identify key bacteria in the cassava rhizosphere that may promote cassava productivity. The identified target bacterial genera will be valuable for species-level identification and for developing biofertilizers containing a range of microorganisms capable of nitrogen fixation, potassium and phosphate solubilization, plant hormone production, and stress resistance. Ultimately, this research provides insights that can guide safer, more cost-effective, and sustainable farming practices.

## MATERIALS AND METHODS

### Experimental sites

Cassava was grown at two sites in Northeastern Thailand, including Nampong district, Khon Kaen Province and Seungsang district, Nakhon Ratchasima Province, both of which are owned by smallholder farmers who granted permission to sample soil on their properties. The Northeastern region of Thailand is the current dominant cassava growing region of the country. The soil type at the Nampong site (Latitude: 16.823750, Longitude: 102.979556) is a Nam Phong soil series (loamy, siliceous, isohyperthermic Grossarenic Haplustalfs) and the soil at the experimental field has a sandy texture (*Kunlanit, Khwanchum & Vityakon, 2020*). The soil type at the Seungsang site (Latitude: 14.490722, Longitude: 102.508167) is a Chokchai soil series (very fine, kaolinitic, isohyperthermic Rhodic Kandiustox) and the experimental field soil has a silty clay texture (*Sirichuaychu et al., 2005*). The soil chemical properties and growth season temperatures of the two experimental site regions were explained in our previous study (*Thompson et al., 2025*).

### Experimental treatments

The investigation of changes in the rhizosphere microbiome under various fertilizer applications was conducted across eight experimental treatments, using the commercial Rayong 72 cassava cultivar, which is well-known for its high yield and resistance to drought (*Sarakarn et al., 2000*). The experiment was designed in a randomized complete block design (RCBD) with five replicates. Eight experimental treatments with different fertilizer management protocols consisted of (T1) no fertilizer application (control), (T2) recommended dosages of synthetic fertilizer based on soil analysis (RDCF) (*Department of Agriculture, 2005*), (T3) 3.12 t ha$^{-1}$ of chicken manure (CM), (T4) 937.5 L ha$^{-1}$ of swine manure extract (SME), (T5) 3.12 t ha$^{-1}$ of chicken manure + 937.5 L ha$^{-1}$ of swine manure extract (CM + SME), (T6) 50% of the recommended dose of synthetic fertilizers + 1.56 t ha$^{-1}$ of chicken manure (1/2 RDCF + 1/2 CM), (T7) 50% of the recommended dose of synthetic fertilizers + 468.8 L ha$^{-1}$ of swine manure extract (1/2 RDCF + 1/2 SME), and (T8) 3.12 ton ha$^{-1}$ of chicken manure + stalk soaking with plant growth-promoting rhizobacteria (PGPR-3) solution containing two types of bacteria: *Azospirillum brasilense* and *Gluconacetobacter diazotrophicus* (CM + PGPR). Details of application rate and macronutrient contents for each fertilizer application were explained in our previous study (*Thompson et al., 2025*).

### Plant materials, DNA extraction and 16S rRNA amplicon sequencing

Soil samples from cassava tubers were collected during the growing season at three stages, including 2, 5 and 10 months after planting (MAP) at both Nampong and Seungsang field sites. Soil samples were collected from around the tubers of plants grown with all eight experimental treatments. For each stage, three cassava plants (replicates) were sampled per treatment, resulting in a total of 24 samples per site for each stage (48 samples for both sites at each cassava stage). A total of 144 samples were collected, from three stages of growth (2, 5, and 10 MAP) under the eight nutrient management treatments, at both study sites. For each sample, soil surrounding the tubers was collected by gently shaking and manually

removing it with gloved hands. The soil of each sample was collected in plastic bags and was thoroughly mixed by repeatedly flipping the bags. For DNA extraction, 250 mg of soil from each sample was used per tube. The soil samples were stored at −20 °C before extraction. DNA extraction was carried out using the DNeasy PowerSoil Pro kit (Qiagen, Hilden, Germany). DNA quality and concentration were assessed using both 1% agarose gel and a Nanodrop ND-1000 Spectrophotometer. The extracted DNA was stored at −20 °C before proceeding to the sequencing step. For amplicon metagenomic sequencing, 200 ng of total DNA and more than 10 ng/µl of the DNA concentration was needed. The 16S rRNA amplicon amplification targeted the V3–V4 regions, with a fragment length of 470 bp. The primers used to amplify these regions were primer 341F, 5′CCTAYGGGRBGCASCAG 3′ and primer 806R, 5′ GGACTACNNGGGT ATCTAAT 3′. The amplicon sequencing was conducted by an Illumina NovaSeq 6000 platform by Novogene Corporation (Singapore) using 2 × 250 bp pair-end reads. The output of total 16S rRNA amplicon sequences was used from microbiome analysis in the next step.

## Microbiome analysis

After the Illumina sequencing step, all reads of the 16S rRNA amplicon sequences from all 144 soil samples were prepared for the microbiome analysis. Quality control for raw paired-end sequences was performed by trimming adapter and primer sequences and filtering low quality sequences (Phred Quality Score < 30) using the FASTP program (*Chen et al., 2018*). Cleaned paired-end sequences were subsequently joined into single-end sequences using FLASH version V1.2.7 (*Magoč & Salzberg, 2011*). Data analysis of 16S rRNA sequences was performed using QIIME 2 (version 2022.02) (*Bolyen et al., 2019*). The 16S rRNA sequences were quality filtered, denoised, and amplicon sequence variants (ASVs) were constructed using DADA2 (version 1.14) (*Callahan et al., 2016*) implemented in QIIME2. The minimum total frequency and number of observed sample parameters were defined as 100 and 2, respectively. Taxonomic classification was performed using the BLAST+ consensus taxonomy classifier against the SILVA database (released version 138.1) (*Quast et al., 2013*) by specifying the following parameters: percent identity cutoff = 0.9 and minimum fraction of consensus assignment = 0.5. Any sequences containing chloroplast or mitochondria according to taxonomic classification were removed before moving to the next step. Beta diversity analysis based on Bray-Curtis dissimilarity matrix was used to measure the similarity among samples from different treatments and was visualized by principal coordinate analysis (PCoA). The significant effects of treatments in shaping bacterial communities were calculated by permutational multivariate analysis of variance (PERMANOVA) using 999 as the number of permutations. The raw sequence data were deposited under NCBI BioProject Accession number PRJNA1202398. Due to the limitations of using only 16S rRNA amplicon data, which does not capture functional microbial potential, FAPROTAX (*Louca, Parfrey & Doebeli, 2016*) analysis was perform to translate our bacterial taxonomic profiles into putative metabolic functions, allowing us to infer the ecological roles of bacterial community across treatments.

## Analysis of predominant bacteria and correlation coefficient with cassava production

The predominant bacteria were identified by comparing genus-level bacteria that showed significant differences across experimental treatments at the three stages (2, 5, and 10 MAP). Emphasis was placed on experimental treatments that contributed to the highest growth and yield of cassava at the Nampong and Seungsang sites. Comparisons were made using White's non-parametric $t$-test in STAMP (*Parks et al., 2014*) with a significance level of $p < 0.05$. White's non-parametric $t$-test was selected for its robustness in detecting differences in bacterial abundance in microbiome data, accommodating the non-normal distributions and small sample sizes. To identify the predominant bacteria at genus-level correlated with fresh tuber yield and starch content recorded at 10 MAP, the Spearman correlation coefficient was used with Scipy python package (*Virtanen et al., 2020*). Details of the fresh tuber yield and starch content for the experimental treatments are reported in a previous study (*Thompson et al., 2025*).

## RESULTS

### DNA extraction and 16S rRNA sequencing

DNA from a total of 144 soil samples, from the eight treatments at the Nampong and Seungsang sites, collected at three cassava stages (2, 5, and 10 MAP), was extracted and used for 16S rRNA amplicon sequencing using Illumina sequencing technology. The quality of the genomic DNA was found to be high and sufficient for sequencing. The genomic DNA bands from all soil samples were larger than 5,000 bp, with the 16S rRNA fragment approximately 1,500 bp in size. The DNA concentration in nearly all soil samples exceeded 50 ng/μl. The cassava plants, along with their roots and tubers with surrounding soil, collected at 2, 5, and 10 MAP from the Nampong and Seungsang sites, is shown in Fig. 1, along with the DNA patterns of soil samples.

### Amplicon sequence variant derived from 16S rRNA sequences

To study the cassava rhizosphere bacterial communities resulting from different fertilizer treatments, 16S rRNA amplicon sequencing was performed using the Illumina NovaSeq 6000 platform with paired-end 250 bp reads. The amplicons targeted the V3–V4 variable regions, with a fragment size of 470 bp. After sequencing DNA from the 144 soil samples, the initial analysis using MultiQC showed that the read lengths of all 144 samples ranged from 386 to 430 bp. Amplicon sequence variant (ASV) data for cassava rhizosphere soil at 2, 5, and 10 MAP at the Nampong site is shown in Table S1. At 2 MAP, the non-chimeric ASV count ranged from 84,533 to 113,786, accounting for 43.76% to 60.27% of the input 16S rRNA sequences. At 5 MAP, the non-chimeric ASV count ranged from 89,748–156,981, accounting for 69.16–84.02% of the input 16S rRNA sequences. At 10 MAP, the non-chimeric ASV count ranged from 69,497–161,325, accounting for 70.99–87.03% of the total input 16S rRNA sequences. ASV data for cassava rhizosphere soil at 2, 5, and 10 MAP at the Seungsang site is shown in Table S2. At 2 MAP, the non-chimeric ASV count ranges from 78,003–100,871, accounting for 42.42–51.7% of the

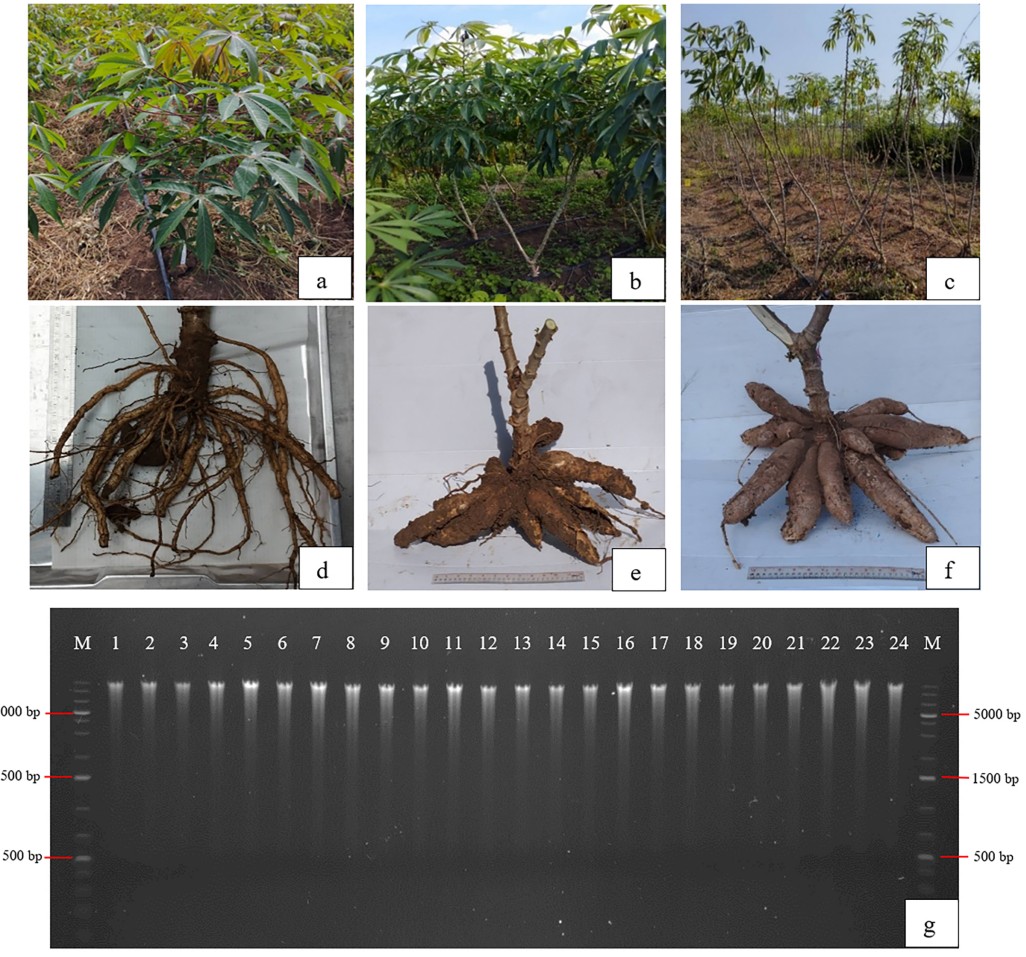

**Figure 1** Examples of cassava stem and tuber characteristics from which soil around the roots was collected for microbiome study at 2 (A, D), 5 (B, E), and 10 MAP (C, F) and DNA patterns of soil samples around the cassava tubers (G). M = GeneRuler$^{TM}$ 1 kb Plus DNA ladder.

total input 16S rRNA sequences. At 5 MAP, non-chimeric ASV count ranged from 91,829–143,529, accounting for 73.92–84.49% of the total input 16S rRNA sequences. At 10 MAP, non-chimeric ASV count ranged from 54,038–145,551, accounting for 41.75–85.68% of the total the input 16S rRNA sequences.

## Taxonomic classification at phylum and genus levels

The microbial composition at the phylum level for all cassava rhizosphere soil samples at 2, 5, and 10 MAP from the Nampong and Seungsang sites showed similar proportions, as illustrated in Fig. 2A. It was found that the microbial populations at the phylum level at 2 MAP were noticeably different from those at 5 and 10 MAP, with the populations at 5 and 10 MAP showing only slight differences. At 2 MAP, the Firmicutes phylum was the most abundant, followed by Proteobacteria and Actinobacteria. At 5 and 10 MAP,

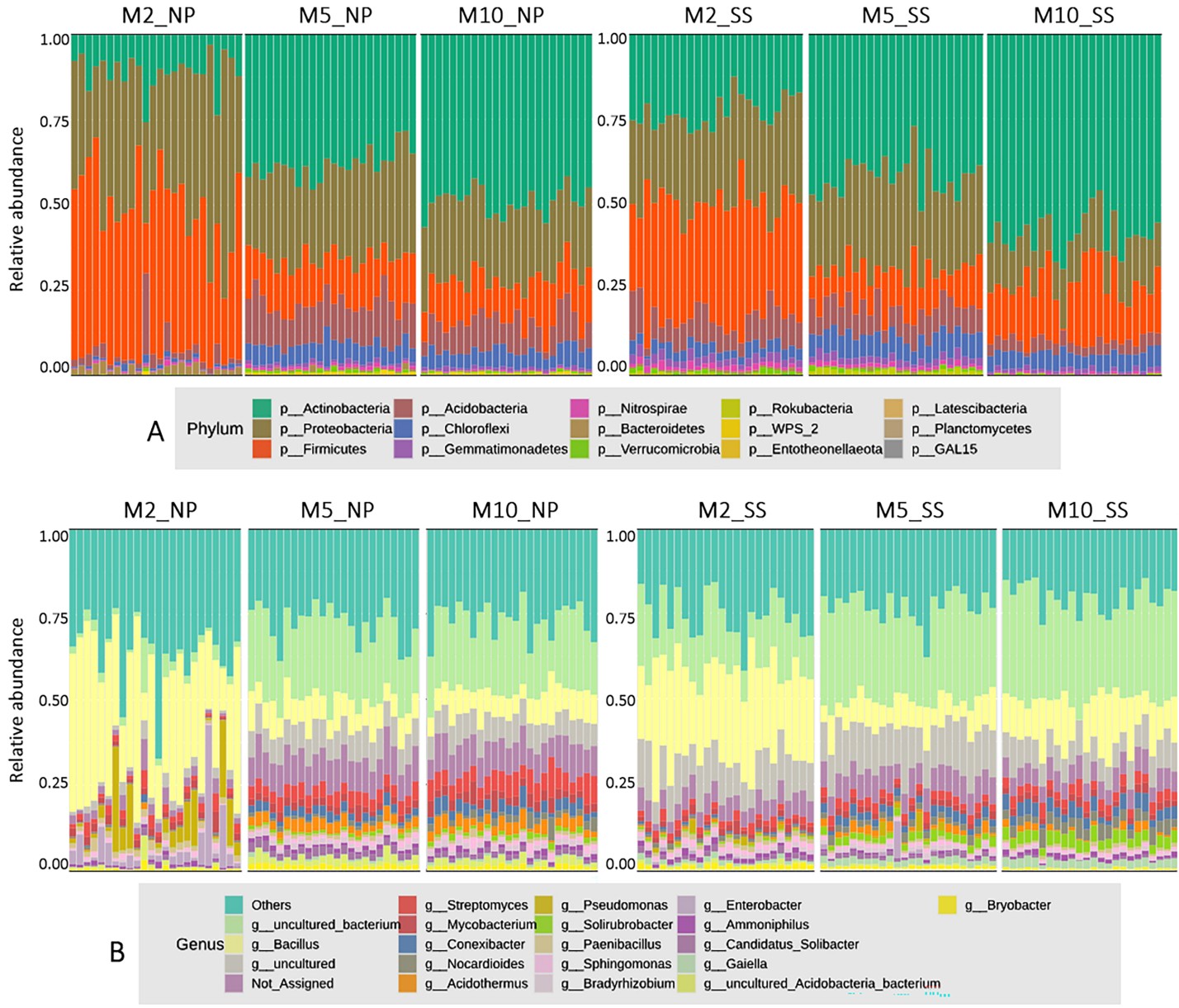

**Figure 2** The relative abundances of microorganisms at the phylum (A) and genus (B) levels in soil samples from the cassava rhizosphere from Nampong (NP) and Seungsang sites, collected at 2 (M2_NP, M2_SS), 5 (M5_NP, M5_SS), and 10 (M10_NP, M10_SS) MAP.

Actinobacteria was the most abundant phylum, followed by Proteobacteria and Firmicutes.

The analysis of microbial proportions at the genus level for all cassava rhizosphere samples at 2, 5, and 10 MAP showed similar results at both Nampong and Seungsang sites, as illustrated in Fig. 2B. The microbial populations at the genus level at 2 MAP differed from those at 5 and 10 MAP, mirroring the differences observed at the phylum level. The microbial populations at the genus level were similar at both 5 and 10 MAP. At 2 MAP, the

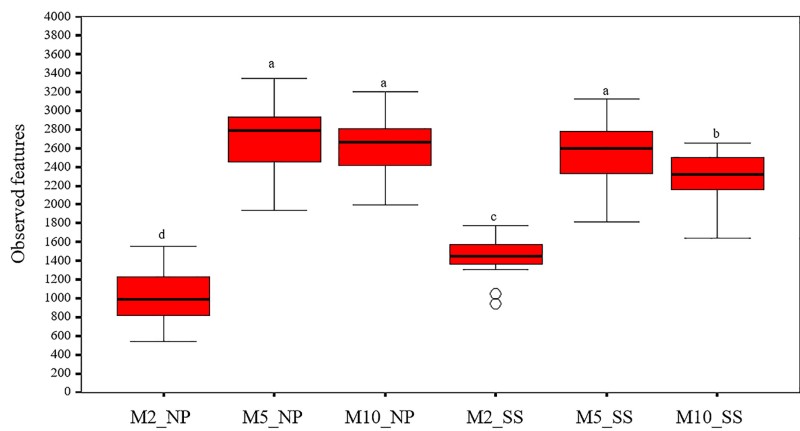

**Figure 3 Comparison of alpha diversity in the cassava rhizosphere from Nampong (NP) and Seungsang (SS) at 2 (M2_NP, M2_SS), 5 (M5_NP, M5_SS), and 10 (M10_NP, M10_SS) months after planting (MAP).** The different letters indicate significant mean differences based on the Kruskal-Wallis multiple comparison test (pairwise) at *p* value < 0.05.

*Bacillus* genus was the most abundant. However, at 5 and 10 MAP, *Bacillus* significantly decreased, though it remained the most abundant genus compared to others, including *Steptomyces*, *Mycobacterium*, *Conexibacter* and *Acidothermus*.

## Alpha diversity of microorganisms

The alpha diversity indicated that overall observed microbial features in the soil around the tubers at 2, 5, and 10 MAP were similar at both the Nampong and Seungsang sites (Table S3, Fig. 3). The highest abundance of observed microbial features occurred at 5 MAP, with a decline at 10 MAP. The number of observed features at 5 and 10 MAP was significantly higher than that observed at 2 MAP at both sites. At Nampong, observed microbial features decreased at 10 MAP, but no significant difference was found between 5 and 10 MAP. In contrast, at Seungsang, observed microbial features decreased at 10 MAP and showed a significant difference compared to 5 MAP.

The observed bacterial features around the tubers at both sites for each treatment at 2, 5, and 10 MAP are illustrated on Fig. 4 and the details from treatments that resulted in significant differences are shown in Table S4. At Nampong, no significant differences in observed microbial features were found across experiments at 2 MAP. However, by 5 MAP, the no-fertilizer control (T1) had fewer observed microbial features than treatments that included chicken manure, such as CM (T3), CM + SME (T5), 1/2 RDCF + 1/2 CM (T6), and CM + PGPR (T8), while no significant differences were observed among the control (T1), chemical fertilizer (T2), and SME (T4) treatments. At 10 MAP, the alpha diversity of CM (T3) remained significantly higher than that of the control (T1) and RDCF (T2).

At Seungsang, significant differences among treatments were observed as early as 2 MAP, with SME (T4) and CM + SME (T5) treatments showing higher bacterial diversity than the no-fertilizer control (T1) and chemical fertilizer RDCF (T2) treatments. At

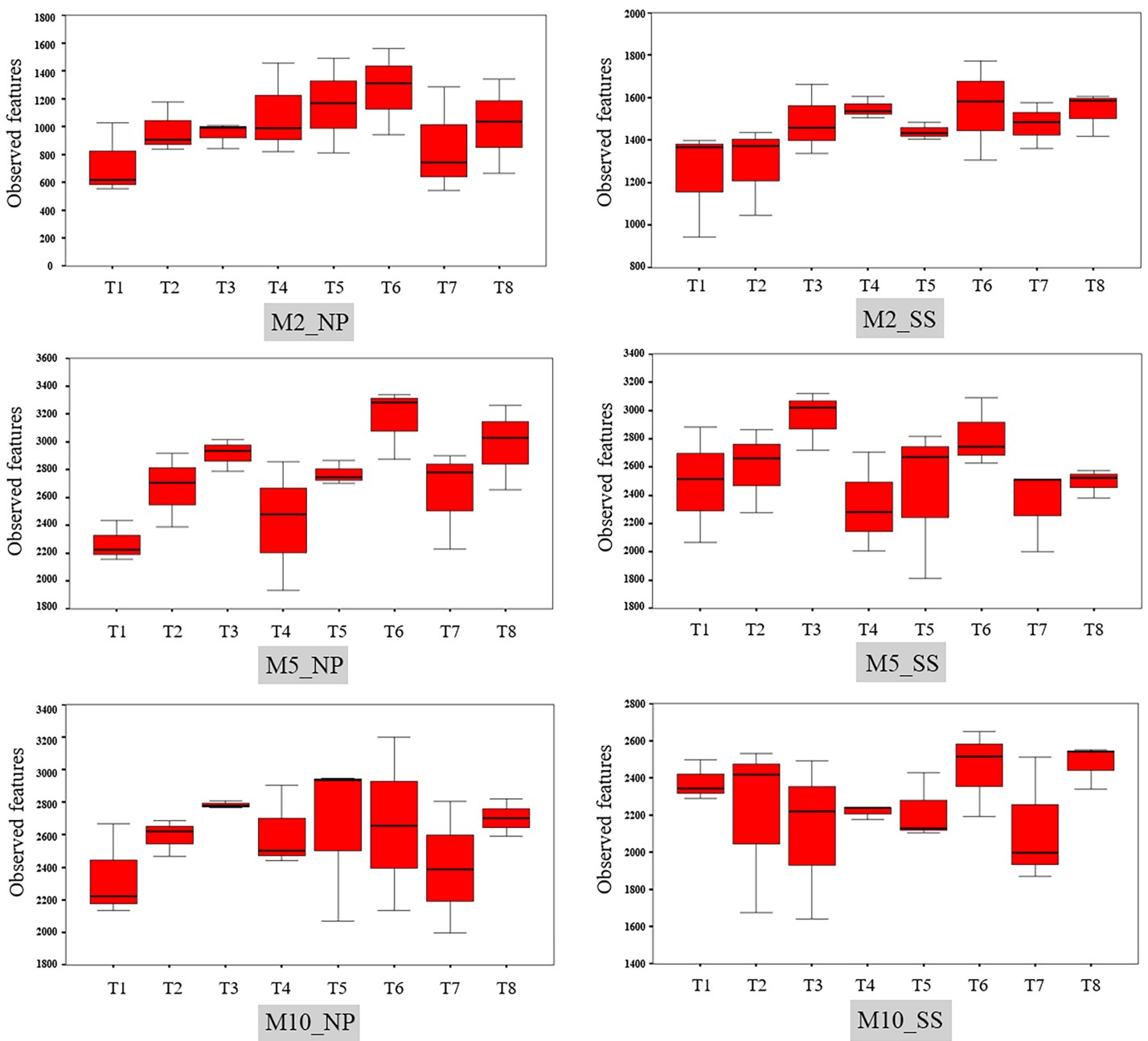

**Figure 4 Comparison of cassava rhizophere alpha diversity resulting from different treatments (T1–T8) in Nampong (NP) and Seungsang (SS) at 2 (M2_NP, M2_SS), 5 (M5_NP, M5_SS), and 10 (M10_NP, M10_SS) months after planting (MAP).**

5 MAP, CM (T3) and 1/2 RDCF + 1/2 CM (T6) treatments exhibited greater alpha diversity than did SME (T4), 1/2 RDCF + 1/2 SME (T7), and CM + PGPR (T8) treatments. At 10 MAP, significant differences were also observed, with the control (T1) and CM + PGPR (T8) treatments displaying higher alpha diversity than what resulted from the SME (T4) treatment.
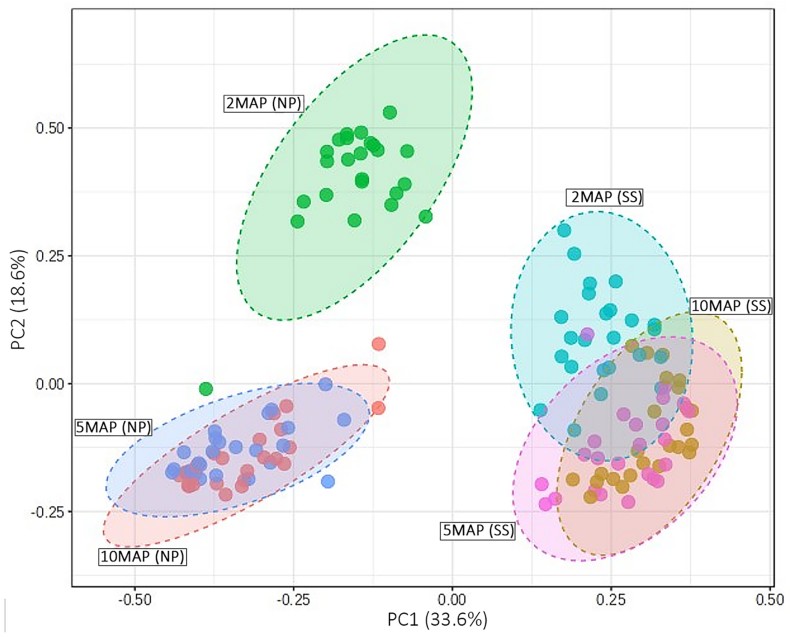

**Figure 5** Comparison of beta diversity using a PCoA plot based on Bray-Curtis (non-phylogenetic diversity metrics were calculated) for microorganisms in the cassava rhizosphere soil at 2, 5, and 10 months after planting (MAP) in Nampong (M2_NP, M5 _NP, and M10_NP, respectively) and Seungsang (M2_SS, M5 _SS, and M10_SS, respectively).

## Beta diversity of microorganisms

The beta diversity of microorganisms in the soil surrounding cassava roots at 2, 5, and 10 MAP in Nampong and Seungsang is illustrated in Fig. 5. The results indicate significant differences in beta diversity across the different growth stages within each site. Additionally, the beta diversity between the two sites also showed slight variations. At Nampong, the beta diversity at 2 MAP differed significantly from that at 5 and 10 MAP, while the diversity at 5 and 10 MAP was nearly the same. Similarly, at Seungsang, there was some overlap in beta diversity across 2, 5, and 10 MAP, but the diversity at 5 and 10 MAP was almost identical, as observed in Nampong. However, within each treatment, the beta diversity at 2, 5, and 10 MAP did not show significant differences for both sites (Table S5).

## Analysis of metabolic functions of microorganisms and predominant bacteria resulting from different fertilizer applications

To compare bacterial genera resulting from different treatments, bacteria were classified from soil samples around cassava tubers grown in Nampong and Seungsang at 2, 5, and 10 MAP. At 2 MAP, 95 bacterial genera were identified, accounting for approximately 66.25–96.15% of the total bacteria found in the 48 soil samples collected from both sites. At 5 MAP, 92 bacterial genera were identified, accounting for approximately 56.86–77.34% of the total bacteria found in the 48 soil samples from both sites. At 10 MAP, 66 bacterial genera were identified, accounting for approximately 63.41–81.81% of the total bacteria found in the 48 soil samples from both sites. Details of bacterial genera are provided in

**Table 1** Target bacteria that are likely to promote growth and yield in cassava at 2, 5, and 10 MAP.

| Cassava stages | Target bacteria genera | |
| --- | --- | --- |
| | **Nampong site** | **Seungsang site** |
| 2 MAP | *Pseudomonas* | 1) *Tumebacillus* <br> 2) *Lysinibacillus* <br> 3) *Paenibacillus* |
| 5 MAP | 1) *Paenibacillus* <br> 2) *Dongia* <br> 3) *Acidibacter* <br> 4) *Spingomonas* | *Bacillus* |
| 10 MAP (age at harvesting) | *Bacillus* | |

Table S5. To infer putative metabolic functions from bacterial taxonomic profiles, we employed FAPROTAX to assess the ecological roles of bacterial communities across treatments, as presented in Fig. S1 and Table S6. The dominant predicted metabolic functions were chemoheterotrophy and aerobic chemoheterotrophy, observed at 2, 5, and 10 MAP. These functions peaked at 10 MAP at both the Nampong and Seungsang sites. However, there were no clear differences in the predicted metabolic functions among the different treatments.

Previous work (*Thompson et al., 2025*) has shown that CM (T3), CM + SME (T5), and CM + PGPR (T8) treatments had a significant impact on cassava growth and yield. Specifically, T3 and T5 had a pronounced effect at Seungsang, while T8 notably influenced cassava growth and yield at Nampong. Therefore, the predominant bacteria in treatments T3, T5, and T8 were considered to be target bacteria. In summary, these target bacteria, including *Pseudomonas*, *Tumebacillus*, *Lysinibacillus*, *Paenibacillus*, *Dongia*, *Acidibacter*, *Sphingomonas*, and *Bacillus* are likely to contribute to cassava productivity at 2, 5, and 10 MAP. A comparison of dominant bacteria, analyzed using White's non-parametric *t*-test in STAMP, is presented in Table 1 and Fig. 6.

## Correlation of predominant bacteria with cassava production

Correlation analysis was conducted to evaluate the relationship between target bacteria and cassava fresh tuber yield as well as between target bacteria and starch content at harvesting time, 10 MAP. Of the eight targeted genera, only six were identified at 10 MAP, including *Tumebacillus*, *Bacillus*, *Paenibacillus*, *Lysinibacillus*, *Dongia*, and *Sphingomonas*. Spearman correlation analysis indicated that *Tumebacillus* was the most prominent genera, showing a significant correlation with fresh tuber yield ($p = 0.041$ at Seungsang and $p = 0.060$ at Nampong) (Fig. 7, Table 2). *Paenibacillus* and *Bacillus* also exhibited a notable relationship with yield ($p = 0.054$ at Seungsang and $p = 0.062$ at Nampong) (Fig. 7, Table 2). However, Spearman correlation analysis found no significant relationship between the six target bacteria and starch content (Table S8).

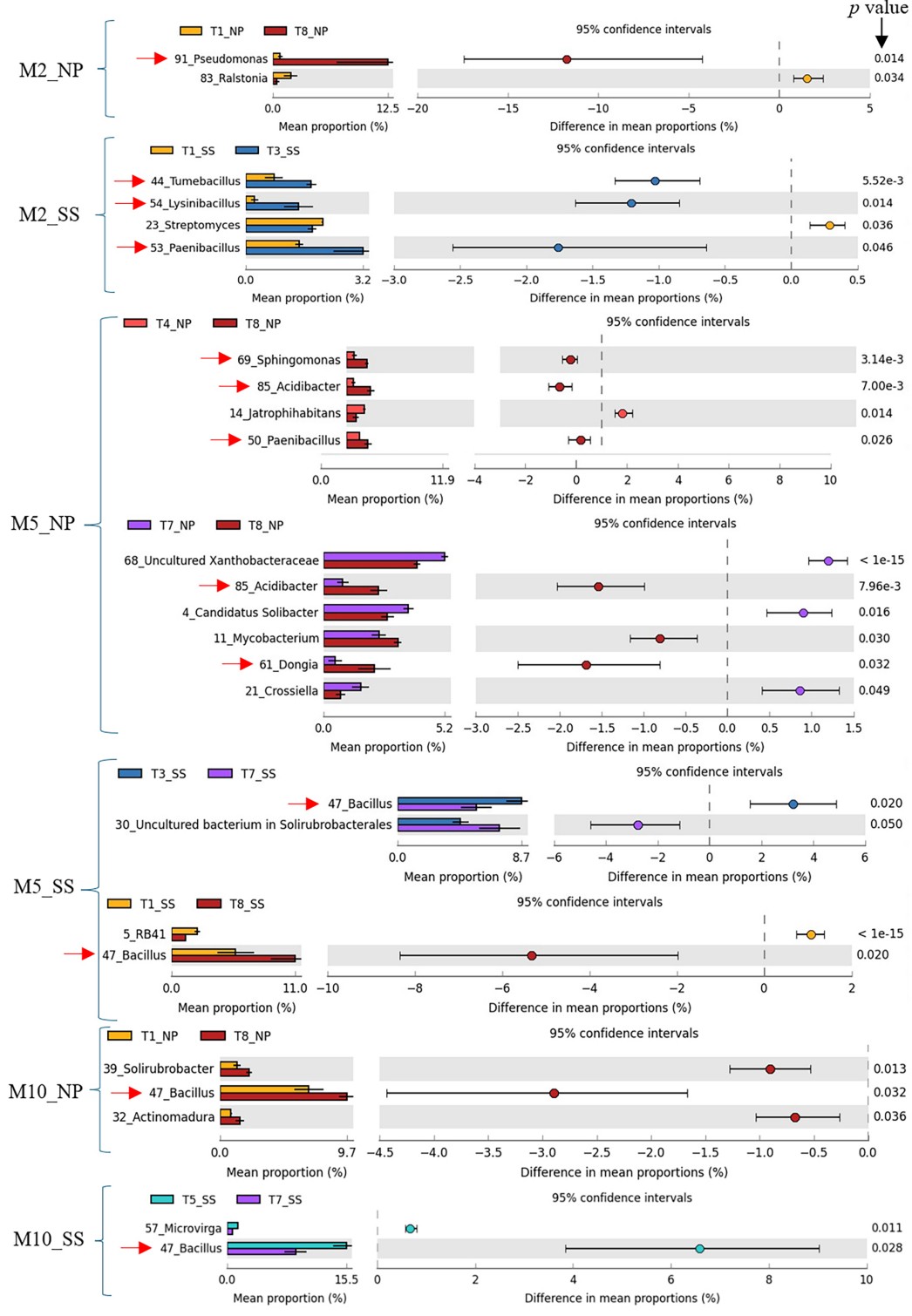

**Figure 6 Target bacteria genera in the cassava rhizosphere were identified by comparing T3, T8, and T5 with other experimental treatments, including T1, T4, and T7, using White's non-parametric t-test in STAMP, at Nampong and Seungsang at 2 (M2_NP, M2_SS), 5 (M5_NP, M5_SS) and 10 (M10_NP, M10_SS) months after planting (MAP).** The target bacteria genera were pointed by red arrows.

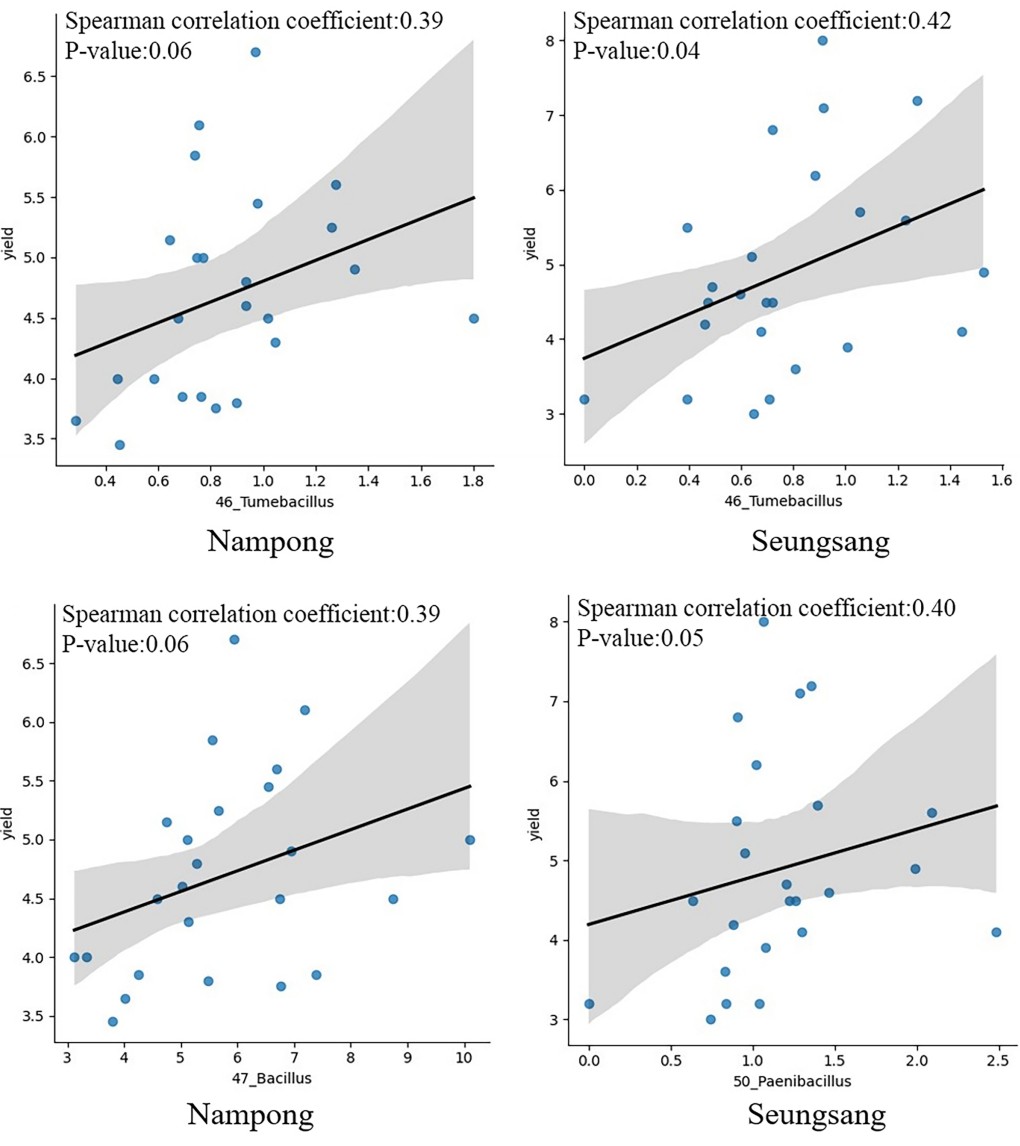

**Figure 7 The Spearman correlation coefficient graph of the target bacterial genera, including *Tumebacillus*, *Bacillus* and *Paenibacillus* that were most associated with yield at the harvest stage of 10 MAP, grown in Nampong and Seungsang.**

## DISCUSSION

This study focuses solely on the soil surrounding cassava tubers. The findings indicate that the proportion of microorganisms at both phylum and genus levels in all soil samples around cassava tubers at 2, 5, and 10 MAP in Nampong and Seungsang followed the same pattern. At the phylum level, Firmicutes was the most abundant at 2 MAP, followed by Proteobacteria and Actinobacteria. However, at 5 and 10 MAP, Actinobacteria was the most dominant, followed by Proteobacteria and Firmicutes. These findings correspond with previous research, which identified Actinobacteria, Proteobacteria, and Firmicutes as

**Table 2 Results of the Spearman correlation coefficient analysis of six target bacterial genera associated with yield at the harvest stage of 10 MAP grown in Nampong and Seungsang.**

| sites | Bacteria genus | Value of spearman correlation coefficient | p-value |
|---|---|---|---|
| NP | 46_Tumebacillus | 0.390 | 0.060 |
| NP | 47_Bacillus | 0.387 | 0.062 |
| NP | 50_Paenibacillus | 0.270 | 0.202 |
| NP | 51_Lysinibacillus | 0.143 | 0.504 |
| NP | 54_Dongia | 0.168 | 0.432 |
| NP | 64_Sphingomonas | 0.175 | 0.414 |
| SS | 46_Tumebacillus | 0.420 | 0.041 |
| SS | 47_Bacillus | −0.017 | 0.937 |
| SS | 50_Paenibacillus | 0.398 | 0.054 |
| SS | 51_Lysinibacillus | 0.130 | 0.544 |
| SS | 54_Dongia | −0.069 | 0.748 |
| SS | 64_Sphingomonas | 0.321 | 0.126 |

**Note:**
NP, Nampong; SS, Seungsang.

the dominant bacteria phyla in all samples from the cassava rhizosphere (*Ha et al., 2021*). At the genus level, 2 MAP, the *Bacillus* genus was the most abundant. At 5 MAP and 10 MAP, its numbers significantly decreased but it remained the most abundant genus compared to others, including *Steptomyces*, *Mycobacterium*, *Conexibacter* and *Acidothermus*. *Bacillus* and *Conexibacter* were among the top 10 most abundant bacterial genera in the rhizosphere soil of different cassava genotypes (*Ha et al., 2021*). Additionally, *Bacillus* was reported to be the most abundant genus found within the tuberous roots of different cassava genotypes. Alpha diversity analysis, based on observed microbial features in the soil surrounding tubers at 2, 5, and 10 MAP at both Nampong and Seungsang sites, followed a similar direction. There was a significant increase in alpha diversity of bacteria in the soil around cassava tubers at 5 MAP compared to 2 MAP. However, at 10 MAP, alpha diversity slightly decreased compared to 5 MAP. Comparison of alpha diversity between the treatments revealed differences between the Nampong and Seungsang sites. At Nampong, no significant differences were observed between the treatments at 2 MAP. However, at Seungsang, the treatments resulted in notable differences at 2 MAP. This suggests that the response to fertilizer occurred more quickly at Seungsang than at Nampong, which is reflected in the higher alpha diversity at Seungsang compared to Nampong at 2 MAP. However, at Nampong, alpha diversity increased rapidly, indicating a strong response to fertilizer when the cassava was 5 MAP. The treatments with CM and CM combinations, including T3, T5, T6, and T8, resulted in significantly higher alpha diversity compared to the control (no fertilizer, T1) and RDCF (T2) treatments. At Seungsang, a response to fertilizer was also observed when the cassava reached 5 MAP. It was found that CM (T3) and 1/2 RDCF + 1/2 CM (T6) resulted in higher alpha diversity compared to SME (T4), 1/2 RDCF + 1/2 SME (T7), and CM + PGPR (T8) treatments. At

10 MAP, no significant differences in observed microbial features were found between the control (T1), synthetic fertilizer (T2), and SME (T4) treatments. At both Nampong and Seungsang sites, there was a notable reduction in the observed microbial feature differences resulting from different treatments. However, at 10 MAP, cassava rhizosphere soils resulting from the CM (T3) treatment maintained significantly higher alpha diversity when compared to the control (T1) and RDCF (T2) treatments at the Nampong site. At the Seungsang site, the control (T1) and CM + PGPR (T8) treatments exhibited higher alpha diversity than what resulted from the SME (T4) treatment at 10 MAP.

Analysis of the beta diversity of microorganisms in the soil around the tubers when the cassava was 2, 5, and 10 MAP found that there was a trend consistent with the alpha diversity. The differences in beta diversity at 2, 5, and 10 MAP were likely influenced by seasonal variations. At 2 MAP, cassava was at the beginning of the rainy season. By 5 MAP, the cassava was in the midst of the rainy season, with continuous rainfall encouraging an increase in both the number and diversity of microorganisms. By 10 MAP, the dry season had set in, leading to a decline in both the number and diversity of microorganisms. However, within each treatment, the beta diversity at 2, 5, and 10 MAP did not show significant differences. This data indicates that the response to different fertilizers affects the number and species (alpha diversity) without influencing the overall species diversity (beta diversity).

The CM (T3) and CM + PGPR (T8) treatments, which had the most significant impact on cassava growth and yield, also resulted in the greatest diversity alpha diversity. The predominant genera found in rhizosphere soils resulting from T3 and T8 were, therefore, considered as target bacteria. In summary, the target bacteria likely involved in promoting growth and yield at 2, 5, and 10 MAP, at both sites, include eight bacterial genera: *Pseudomonas*, *Tumbacillus*, *Lysinibacillus*, *Paenibacillus*, *Dongia*, *Acidibacter*, *Sphingomonas*, and *Bacillus*. *Bacillus*, *Sphingomonas*, and *Pseudomonas* were among the top 10 most abundant bacterial genera found within the tuberous roots and among the top 50 found in the rhizosphere soil of different cassava genotypes (*Ha et al., 2021*). In addition, *Pseudomonas*, *Paenibacillus* and *Bacillus* were reported to be part of the core microbial genera in the cassava rhizosphere (*Zhang et al., 2021*). Spearman correlation coefficient analysis revealed that the *Tumbacillus* genus significantly correlated with yield ($p$ value = 0.041 at Seungsang and $p$ value = 0.060 at Nampong). However, further functional assays are needed to evaluate their contribution to cassava yield. *Paenibacillus* and *Bacillus* also exhibited notable but non-significant relationships with yield, with $p$ values of 0.054 at Seungsang and 0.062 at Nampong, respectively. Some of these target bacteria have been reported to be involved in soil improvement and plant growth in previous studies. *Streptomyces*, *Acidibacter*, *Steroidobacter*, *Sphingomonas*, and *Pseudonocardia* are key genera in the plant microbiome and are associated with the minerals N, P, and K, as well as with soil organic matter (*Vimal et al., 2024*). The *Pseudomonas* genus is involved in nitrogen fixation in roots, similar to other bacteria, such as *Azospirillum*, *Enterobacter*, and *Klebsiella*, which all promote plant growth

(*Hayat et al., 2010*). In this current study, treatment with CM + PGPR-3, which contains *Azospirillum brasilense*, may enhance nitrogen fixation in the rhizosphere of the cassava plant. The *Bacillus* genus is a group of bacteria that promotes plant growth and development by releasing plant hormones (*Shahzad et al., 2021*). For example, the *B. subtilis* strain B4 increases IAA levels to support plant growth. Additionally, *Bacillus* and *Paenibacillus* contribute to phosphate dissolution, making it more available to plants (*Hayat et al., 2010*). *Tumbacillus* has been found to possess 5–15 biosynthetic gene clusters (BGCs), which may serve as potential sources for producing natural products (*Kikuchi et al., 2023*). *Sphingomonas* can promote plant growth by producing IAA hormones and enhancing resistance to stress conditions (*Lombardino et al., 2022*). *Lysinibacillus* is likely to support plant growth and recovery, as it may serve as a source of antibacterial, antifungal, and biopesticidal compounds (*Jamal & Ahmad, 2022*). Some of these target bacteria are among those that help improve soil by processes such as phosphate dissolution, including *Enterobacter agglomerans*, *Pseudomonas* spp., and *Achromobacter* spp. They are also among those reported to produce IAA hormones that contribute to plant growth, such as *Pantoea agglomerans*, *Burkholderia* spp., *Bacillus* spp., and *Pseudomonas* spp. Furthermore, they help inhibit fungal pathogens by producing surfactins (*Paenibacillus* spp.) and pumilacidins (*Bacillus pumilus*) (*Frediansyah, 2021*). Parallel changes in soil organic matter, nutrient status, or pH may influence the bacterial communities differently at the two sites. However, further investigation is needed to understand the specific effects.

## CONCLUSIONS

Since CM and CM + PGPR treatments had a greater impact on bacterial diversity (both abundance and species composition) compared to the control (no fertilizer) and RDCF treatments, and because they significantly influenced cassava growth and yield, the predominant rhizosphere bacteria resulting from these treatments were considered as target bacteria. In summary, the target bacteria likely associated with promotion of cassava growth and yield at 2, 5, and 10 MAP, across both sites, included eight genera: *Pseudomonas*, *Tumebacillus*, *Lysinibacillus*, *Paenibacillus*, *Dongia*, *Acidibacter*, *Sphingomonas*, and *Bacillus*. Spearman correlation coefficient analysis revealed that *Tumebacillus* was the prominent bacteria genus most significantly associated with yield, with a $p$ value of 0.041 at Seungsang. This bacterial data will serve as a key target for developing biofertilizers enriched with beneficial microorganisms in the future. Utilizing beneficial microorganisms can enhance soil fertility, boost nutrient availability, and improve resistance to diseases and pests, ultimately increasing crop yields. Effective soil management, achieved by promoting beneficial microorganisms in the rhizosphere, can also reduce reliance on chemical inputs such as fertilizers, insecticides, and herbicides.

## ACKNOWLEDGEMENTS

We would like to thank Dr. Chutintorn Yundaeng for research project support.

### Funding

This work was supported by the National Research Council of Thailand (NRCT) (No. N21A660520). The funders had no role in study design, data collection and analysis, decision to publish, or preparation of the manuscript.

### Grant Disclosures

The following grant information was disclosed by the authors:
National Research Council of Thailand (NRCT): N21A660520.

### Competing Interests

The authors declare that they have no competing interests.

### Author Contributions

- Suthasinee Somyong conceived and designed the experiments, performed the experiments, analyzed the data, prepared figures and/or tables, authored or reviewed drafts of the article, and approved the final draft.
- Wuttichai Mhuantong conceived and designed the experiments, performed the experiments, analyzed the data, prepared figures and/or tables, authored or reviewed drafts of the article, and approved the final draft.
- Phakamas Phetchawang performed the experiments, analyzed the data, prepared figures and/or tables, authored or reviewed drafts of the article, and approved the final draft.
- Derrick Keith Thompson conceived and designed the experiments, performed the experiments, analyzed the data, prepared figures and/or tables, authored or reviewed drafts of the article, and approved the final draft.
- Ornprapa Thepsilvisut conceived and designed the experiments, performed the experiments, analyzed the data, prepared figures and/or tables, authored or reviewed drafts of the article, and approved the final draft.
- Wirulda Pootakham conceived and designed the experiments, analyzed the data, prepared figures and/or tables, authored or reviewed drafts of the article, and approved the final draft.

### Field Study Permissions

The following information was supplied relating to field study approvals (*i.e.*, approving body and any reference numbers):
Soil samples were collected from the field of small farm holders.

### Data Availability

The raw sequence data are available at NCBI BioProject: PRJNA1202398

## Supplemental Information

Supplemental information for this article can be found online at http://dx.doi.org/10.7717/peerj.20085#supplemental-information.

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
