# Peer review of "Influence of organic, synthetic and biofertilizers on the diversity of cassava rhizosphere microbiome in Northeastern Thailand"

_PeerJ, doi:10.7717/peerj.20085_

## Round 0.1 · original submission · Major Revisions

Please revise your manuscript by following the reviewers' comments.

**Language Note:** The review process has identified that the English language must be improved. PeerJ can provide language editing services - please contact us at [email protected] for pricing (be sure to provide your manuscript number and title). Alternatively, you should make your own arrangements to improve the language quality and provide details in your response letter. – PeerJ Staff

·

Basic reporting

The manuscript is written in clear, professional English and is easy to follow. The introduction provides sufficient background and a clear rationale for the study, highlighting the importance of soil microbiomes and fertilizer management in cassava cultivation.
Relevant literature is well referenced throughout the manuscript.
The structure adheres to PeerJ standards. Figures and tables are high quality, appropriately labeled, and support the results effectively.
Raw data are available through NCBI (BioProject Accession PRJNA1202398), satisfying journal policy.
Minor recommendation: Consider slight improvements in figure legends, particularly by clarifying axes and group labels in the beta diversity plots.

Experimental design

The research falls well within the scope of the journal and presents original findings.
The study design is sound, involving two field sites, multiple fertilizer treatments, and three cassava growth stages with sufficient replication.
The methods are well described and reproducible. DNA extraction, sequencing, and analysis pipelines are appropriate and detailed.
One suggestion: The manuscript would benefit from a short explanation for the choice of White’s non-parametric t-test for bacterial genus comparisons and whether any multiple testing corrections were applied.

Validity of the findings

The findings are valid and well supported by the data.
The authors link their conclusions clearly to their results, particularly regarding the identification of beneficial bacterial genera associated with cassava growth.
The statistical analysis is appropriate. Reporting effect sizes alongside p-values near the significance threshold (e.g., p = 0.054) would strengthen the biological interpretation of these findings.
It would also be useful to briefly acknowledge the limitations of using only 16S rRNA amplicon data, which does not capture functional microbial potential.

Additional comments

Add brief justification for White’s non-parametric t-test in the methods.
Discuss minor limitations such as seasonal variability and taxonomic resolution (genus-level) in the Discussion section.
Slightly improve figure legends for clarity.
I recommend minor revisions before acceptance.

Reviewer 2 ·

Basic reporting

This manuscript was unambiguous in providing the study's background and hypothesis, with sufficient recent literature references. However, a few grammatical errors need to be improved. My minor concerns are listed below.

Experimental design

This manuscript, titled "Influence of organic, synthetic, and biofertilizers on the diversity of cassava rhizosphere microbiomes in Northeastern Thailand," met the journal's aims and scope. The study clearly aimed to compare changes in microbial communities, particularly bacteria, in terms of microbial diversity and composition, in the rhizosphere of cassava roots at different cultivation periods, including the beginning, middle, and harvesting stages, under various soil management treatments, using Illumina sequencing of 16S rRNA amplicons. The methods were clearly explained and conducted with well-designed experiments and statistical analysis.

Validity of the findings

Although this study on the influence of organic, synthetic, and biofertilizers on the diversity of cassava rhizosphere microbiomes in Northeastern Thailand was well-conducted with proper evidence, some points need to be improved. The comments on the study's findings are listed below.
Major concerns:
1. As the T3 (CM), T5 (CM+SME), and T8 (CM+PGPR-3) highly impacted cassava growth and yields, the predominant bacteria in T3, T5, and T8 were considered to be the targeted bacteria for further study. In my opinion, it is challenging to compare T8 with T3 and T5 because it will have been interfered with by the abundance of PGPR-3 in T8. Thus, I suggested that the name of the species or genus of plant growth-promoting rhizobacteria (PGPR-3) should be indicated to avoid bias.
2. Why did the authors claim that the targeted genera, including Pseudomonas, Tumebacillus, Lysinibacillus, Paenibacillus, Dongia, Acidibacter, Sphingomonas, and Bacillus, were the beneficial bacteria? What are the criteria that the authors use to define? Please clearly explain.
3. As the authors focused on the beneficial bacteria around the roots of cassava, I recommend the authors use the FAPROTAX, which is a database that maps prokaryotic clades (e.g., genera or species) to established metabolic or other ecologically relevant functions (https://pages.uoregon.edu/slouca/LoucaLab/archive/FAPROTAX/lib/php/index.php), to predict the most essential function of the bacterial community associated with cassava roots. It may help the authors to indicate the bacterial community that promotes or hinders cassava growth and yields.
4. Why did the authors point out only the relationship between the Bacillus group and cassava yields? What about other genera? The authors emphasized that the phylum Actinomycetota has increased at 5 and 10 MAP.
5. To better understand the microbial interaction, I suggested that the microbial correlation should be analyzed to provide the fluctuation of bacteria in the impacted treatment.
Minor concerns:
- line 128: Figure 1 --> Fig. 1
- p should be italicized
- t-test --> t should be italicized
- All taxa's names, e.g., phylum, genus, etc., should be italicized

Reviewer 3 ·

Basic reporting

The manuscript “Inûuence of organic, synthetic and biofertilizers on the diversity of cassava rhizosphere microbiomes in Northeastern Thailand” deals with sustainable cassava production under nutrient-depleting monocropping practices in Thailand which is directly relevant to food security and soil health. The author needs to revise before the final decision is made.
Comments
• The difference in the site condition, including rainfall, temperature and soil pH data between Nampong and Seugsang, was not reported, which might affect microbial dynamics.
• The study mainly focuses on microbial diversity without discussing the parallel changes in soil organic matter, nutrient status or pH. The author mention all these if observed in the material and method section.
• LN-109-120: Please rewrite the content.
• The detailed about DNA extraction, sequencing where 16S rRNA or bioinformatics tools used were not mentioned. Author should provide this information in material and method section.
• The author mentions apha diversity and was discussed in the manuscript. The author should mention the beta diversity data, which is essential to the study. The author should discuss in the Discussion section.
• The significant correlation between the organism Tumebacillus and yield does not confirm the causality without further functional assay.
• Only bacterial communities are discussed in the study. The arbuscular mycorrhizal fungi and other microbes may also affect cassava performance, which is mentioned in the manuscript.
• In the manuscript, it is stated that T3 and t8 improved the yield, but the acutal data is missing which need to be presented/ discussed.
• In the conclusion section author should add future thrust of the present study.
• The quality of Fig. 2 should be improved and of high quality.

Experimental design

See section1

Validity of the findings

See section1

Additional comments

See section1

---

## Round 0.2 · accepted · Accept

The manuscript is now ready for acceptance.

·

Basic reporting

Fine

Experimental design

Good

Validity of the findings

It is fine.

Additional comments

Authors have made amendments as suggested by the reviewers. So, the manuscript in current form can be consider for publication.

Reviewer 2 ·

Basic reporting

The revised manuscript is well-written and easy to follow, using clear and professional English. The introduction effectively establishes the study's relevance by providing extensive background information and a strong justification for its focus on the roles of soil microbiomes and fertilizer management in cassava cultivation. The paper also demonstrates a thorough understanding of the subject matter by citing and referencing relevant literature throughout.

Experimental design

The experimental design in this study is sound, as it evaluates the effects of multiple fertilizer treatments across two distinct field sites, which enhances the generalizability and ecological relevance of the findings. The study also includes a dynamic analysis of the plants at three critical growth stages, and utilizes sufficient replication to ensure the statistical power and validity of the results. Moreover, the authors have performed the analysis according to the comments.

Validity of the findings

The revised findings are robust and sound.

Additional comments

-

Reviewer 3 ·

Basic reporting

The authors made significant changes in the manuscript

Experimental design

The authors address all concerns raised by the reviewer

Validity of the findings

.

Additional comments

.